# Allosteric coupling asymmetry mediates paradoxical activation of BRAF by type II inhibitors

Damien M Rasmussen[1,2], Manny M Semonis[1], Joseph T Greene[1], Joseph M Muretta[2], Andrew R Thompson[2], Silvia Toledo Ramos[1], David D Thomas[2], William CK Pomerantz[3], Tanya S Freedman[1,4,5], Nicholas M Levinson[1,5]*

[1]Department of Pharmacology, University of Minnesota, Minneapolis, United States; [2]Department of Biochemistry, Molecular Biology, and Biophysics, University of Minnesota, Minneapolis, United States; [3]Department of Chemistry, University of Minnesota, Minneapolis, United States; [4]Center for Immunology, University of Minnesota, Minneapolis, United States; [5]Masonic Cancer Center, University of Minnesota, Minneapolis, United States

*For correspondence:
nml@umn.edu

## Abstract

The type II class of RAF inhibitors currently in clinical trials paradoxically activate BRAF at subsaturating concentrations. Activation is mediated by induction of BRAF dimers, but why activation rather than inhibition occurs remains unclear. Using biophysical methods tracking BRAF dimerization and conformation, we built an allosteric model of inhibitor-induced dimerization that resolves the allosteric contributions of inhibitor binding to the two active sites of the dimer, revealing key differences between type I and type II RAF inhibitors. For type II inhibitors the allosteric coupling between inhibitor binding and BRAF dimerization is distributed asymmetrically across the two dimer binding sites, with binding to the first site dominating the allostery. This asymmetry results in efficient and selective induction of dimers with one inhibited and one catalytically active subunit. Our allosteric models quantitatively account for paradoxical activation data measured for 11 RAF inhibitors. Unlike type II inhibitors, type I inhibitors lack allosteric asymmetry and do not activate BRAF homodimers. Finally, NMR data reveal that BRAF homodimers are dynamically asymmetric with only one of the subunits locked in the active αC-in state. This provides a structural mechanism for how binding of only a single αC-in inhibitor molecule can induce potent BRAF dimerization and activation.

## eLife assessment

This elegant study presents **important** findings into how small molecules that were originally developed to inhibit the oncogenic kinase, BRAF, instead trigger activation of this kinase target. **Compelling** and comprehensive evidence supports a new allosteric model to explain the paradoxical activation. This rigorous work will be of great interest to biochemists, structural biologists, and those working on strategies to inhibit kinases in the context of human disease.

## Introduction

The protein kinase BRAF is a central component of the MAPK (RAF-MEK-ERK) signaling pathway and plays a critical role in the regulation of eukaryotic cell growth, proliferation, and survival (*Lavoie and Therrien, 2015*; *Wellbrock et al., 2004*). In quiescent cells, BRAF exists in the cytosol as an

autoinhibited monomer (*Park et al., 2019*; *Martinez Fiesco et al., 2022*; *Thevakumaran et al., 2015*). Upon initiation of growth factor signaling, BRAF monomers are recruited to the plasma membrane by RAS-GTP and activated by homodimerization or by heterodimerization with the additional isoforms ARAF and CRAF (*Rushworth et al., 2006*; *Weber et al., 2001*). Dimerization allosterically activates BRAF by triggering a conformational change of the regulatory αC-helix from an inactive αC-out state to an active αC-in state (*Rajakulendran et al., 2009*). Activated BRAF dimers initiate a phosphorylation cascade in which MEK and ERK are sequentially activated by successive phosphorylation events (*Lavoie and Therrien, 2015*).

Mutations in BRAF play a major role in driving human cancers, most notably in approximately half of melanoma cases (*Davies et al., 2002*). The most prevalent BRAF mutation is V600E (*Owsley et al., 2021*), which confers constitutive kinase activity on BRAF by disrupting critical autoinhibitory interactions, allowing monomeric V600E BRAF to phosphorylate its substrate MEK independently of upstream signaling and dimerization (*Yao et al., 2015*; *Wan et al., 2004*). The FDA-approved inhibitors vemurafenib, dabrafenib, and encorafenib show remarkable initial responses in V600E-driven metastatic melanoma patients due to their effective inhibition of V600E BRAF monomers. However, clinical resistance emerges rapidly and is driven by the formation of mutant BRAF dimers, which these drugs fail to inhibit (*Chapman et al., 2011*; *Hauschild et al., 2012*; *Dummer et al., 2018*). Consequently, mechanisms that promote BRAF dimerization, including receptor tyrosine kinase upregulation, activating RAS mutations, and BRAF splice variants, confer resistance to these inhibitors (*Nazarian et al., 2010*; *Poulikakos et al., 2011*; *Wagle et al., 2011*; *Cook and Cook, 2021*; *Proietti et al., 2020*; *Brummer and McInnes, 2020*). In fact, these inhibitors can paradoxically activate BRAF dimers, leading to elevated MAPK signaling in cells containing dimeric BRAF (*Poulikakos et al., 2010*; *Hatzivassiliou et al., 2010*; *Joseph et al., 2010*). This paradoxical activation triggers the emergence of secondary cutaneous carcinomas in many patients treated with these inhibitors (*Su et al., 2012*).

The failure of the FDA-approved BRAF inhibitors to block BRAF dimers is attributed to negative allostery, in which inhibitors preferentially bind the inactive αC-out state of BRAF and are unable to bind to the active αC-in state adopted by BRAF dimers (*Karoulia et al., 2016*). This discovery prompted the development of a new class of inhibitors that recognize the αC-in state and bypass this negative allostery (*Peng et al., 2015*; *Okaniwa et al., 2013*). Remarkably, despite binding to both subunits of the BRAF dimer in X-ray structures (*Peng et al., 2015*; *Yen et al., 2021*; *Tkacik et al., 2023*) and being reportedly equipotent for both subunits (*Cotto-Rios et al., 2020*), these αC-in inhibitors still induce paradoxical activation of MAPK/ERK signaling in cells (*Poulikakos et al., 2010*; *Yen et al., 2021*; *Cotto-Rios et al., 2020*; *Hall-Jackson et al., 1999*; *Nakamura et al., 2013*; *Lai et al., 2022*). Paradoxical activation by αC-in inhibitors is linked to their ability to induce BRAF dimers (*Lavoie et al., 2013*), but the molecular mechanisms triggering this activation remain elusive. Addressing this question could shed light on the shortcomings of numerous drugs currently undergoing clinical trials (*Eisen et al., 2006*; *Sullivan et al., 2020*) and inspire the development of new RAF inhibitors that do not induce paradoxical activation of the MAPK pathway.

Here, we use biophysical techniques tracking BRAF dimerization, activation, and structural changes in solution, to develop a comprehensive model highlighting the allosteric and thermodynamic mechanisms that underpin paradoxical activation by αC-in RAF inhibitors. We demonstrate that all αC-in inhibitors are allosterically coupled to BRAF dimerization to a remarkable degree, and further highlight fundamental distinctions in the coupling mechanisms between type I and type II αC-in inhibitors that explain key differences in how they induce paradoxical activation.

## Results

### Type II inhibitors drive BRAF dimerization through asymmetric allosteric coupling

We used intermolecular FRET to quantify inhibitor-induced BRAF dimerization in vitro. A previously validated construct of the BRAF kinase domain containing 16 solubilizing mutations (*Thevakumaran et al., 2015*; *Tsai et al., 2008*) (hereafter referred to as 'BRAF') was covalently labeled on K547C with donor (Alexa Fluor 488) or acceptor (Alexa Fluor 568) fluorophores, mixed at equal molar ratios and dispensed into multiwell plates (*Figure 1a* and *Figure 1—figure supplement 1*). We recorded donor and acceptor emission spectra on a fluorescence plate reader (*Schaaf et al., 2017*) and utilized spectral

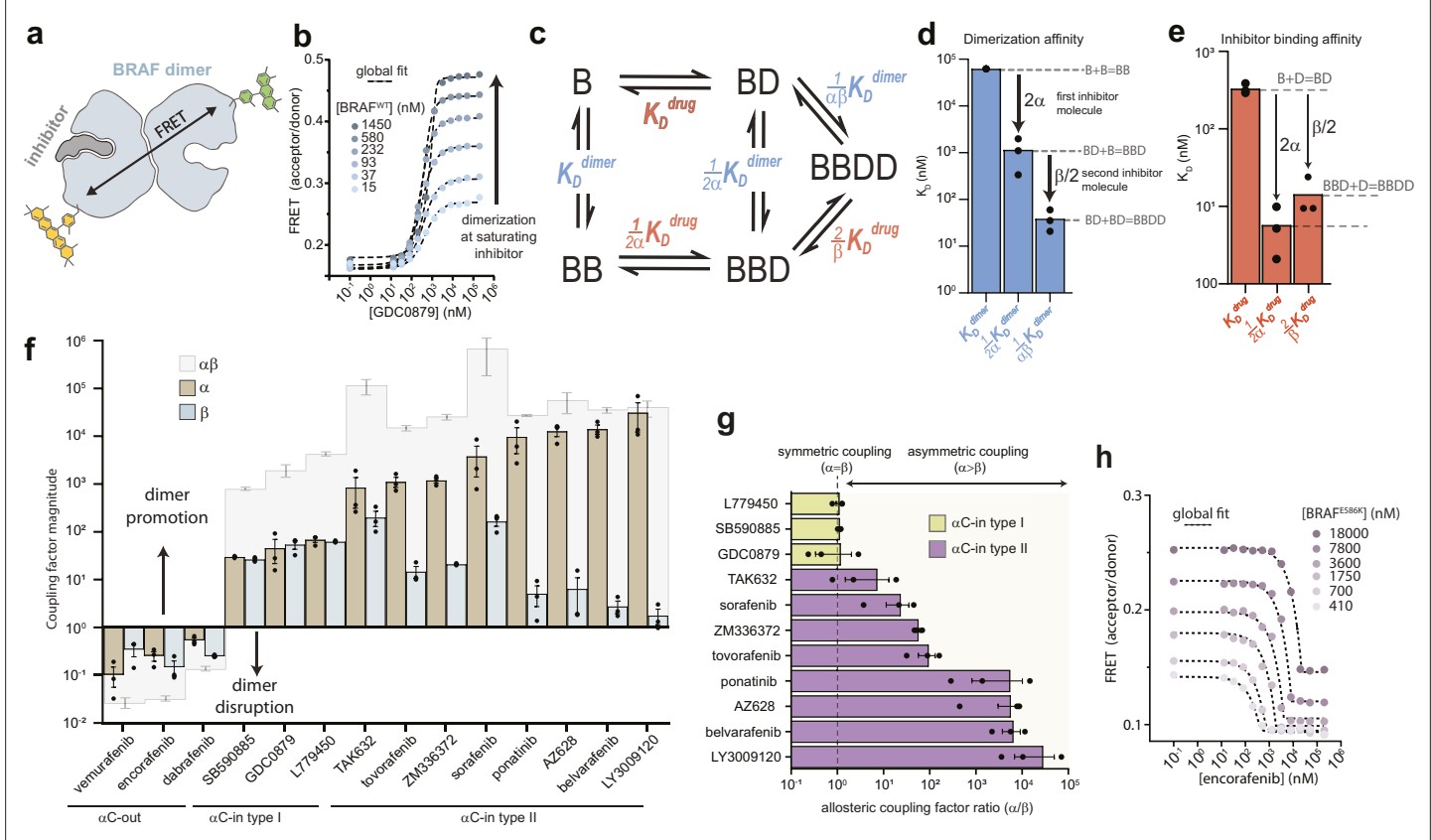

**Figure 1.** Type II αC-in inhibitors drive BRAF dimerization through asymmetric allosteric coupling. (**a**) Schematic of the intermolecular FRET sensor used to quantify BRAF dimerization. A detailed structural model of the FRET sensor is shown in *Figure 1—figure supplement 1*. (**b**) Representative intermolecular FRET experiments measuring BRAF dimerization by type I αC-in inhibitor GDC0879 as a function of BRAF concentration. Dashed lines represent the global fit of these data to the model shown in panel c. Data for all αC-in inhibitors are shown in *Figure 1—figure supplement 3*. (**c**) Model of inhibitor-induced BRAF dimerization used for the global fitting of FRET data in panel b. B represents apo/unbound BRAF monomer, BD drug/inhibitor-bound monomer, BB apo/unbound dimer, BBD dimeric BRAF with one bound inhibitor molecule, and BBDD dimeric BRAF with two inhibitor molecules bound. These biochemical species are linked by the equilibrium dissociation constants described in the main text and methods. (**d, e**) Equilibrium dissociation constants for dimerization (panel d) and inhibitor binding (panel e) determined from global fitting analysis of the GDC0879 experiments shown in panel b. Allosteric coupling factors α and β describe the coupling of BRAF dimerization to the first and second inhibitor binding events, respectively (see Materials and methods), and are similar in magnitude for this type I inhibitor. Dissociation constants for all inhibitors are shown in *Figure 1—figure supplements 5 and 6* . (**f**) Allosteric coupling factors α and β, as well as their product αβ, are shown for all RAF inhibitors examined. Error bars represent the mean ± s.e.m.; n≥3 independent experiments, each performed in duplicate. (**g**) Plots of the allosteric coupling factor ratio α/β for all αC-in inhibitors. (**h**) Representative intermolecular FRET experiments measuring disruption of BRAF$^{E586K}$ dimerization by the αC-out inhibitor encorafenib at increasing BRAF concentrations. Dashed lines represent the global fit to the thermodynamic model shown in panel c.

The online version of this article includes the following source data and figure supplement(s) for figure 1:

**Source data 1.** Tabulated source data for *Figure 1*.

**Figure supplement 1.** X-ray structures showing the BRAF dimer labeled with fluorophores and structural features of type I and type II inhibitor binding modes.

**Figure supplement 1—source data 1.** Tabulated source data.

**Figure supplement 2.** Quantifying apo BRAF dimerization affinity ($K_D^{dimer}$) and dimer induction by GDC0879.

**Figure supplement 2—source data 1.** Tabulated source data.

**Figure supplement 3.** Intermolecular FRET experiments measuring inhibitor-induced BRAF dimerization.

**Figure supplement 3—source data 1.** Tabulated source data.

**Figure supplement 4.** One- and two-dimensional error surfaces show that global fitting yields well-constrained parameters for the allosteric model.

**Figure supplement 4—source data 1.** Tabulated source data.

**Figure supplement 5.** Equilibrium dissociation constants derived from global fitting of FRET data for all type II inhibitors.

*Figure 1 continued on next page*

*Figure 1 continued*

**Figure supplement 5—source data 1.** Tabulated source data.

**Figure supplement 6.** Equilibrium dissociation constants derived from global fitting of FRET data for type I αC-in and αC-out inhibitors.

**Figure supplement 6—source data 1.** Tabulated source data.

**Figure supplement 7.** Inhibitor affinities for monomeric BRAF ($K_D^{drug}$) measured by intramolecular FRET.

**Figure supplement 7—source data 1.** Tabulated source data.

**Figure supplement 8.** The A481F active site mutation blocks inhibitor binding and prevents inhibitor-induced dimerization.

**Figure supplement 8—source data 1.** Tabulated source data.

**Figure supplement 9.** The binding of a single αC-in type II molecule to the BRAF dimer is sufficient to dramatically increase dimerization affinity.

**Figure supplement 9—source data 1.** Tabulated source data.

**Figure supplement 10.** Intermolecular FRET experiments measuring the disruption of BRAF$^{E586K}$ dimerization by αC-out inhibitors.

**Figure supplement 10—source data 1.** Tabulated source data.

**Figure supplement 11.** Only asymmetric allosteric models correspond to realistic levels of catalytic activity of BBD dimers for type II inhibitors.

**Figure supplement 11—source data 1.** Tabulated source data.

deconvolution to obtain accurate values for the acceptor/donor (A/D) ratio, from which dimerization was inferred (*Figure 1b*). In control experiments without drug, the baseline dimerization affinity of apo BRAF ($K_D^{dimer}$) was found to be 62.4±2.9 μM, in agreement with previously reported values (*Lavoie et al., 2013*; *Figure 1—figure supplement 2a and b*). We then examined BRAF dimerization in the presence of a diverse set of 11 αC-in (3 type I, 8 type II) and 3 αC-out RAF inhibitors (*Supplementary file 1*). All αC-in inhibitors induced large increases in the A/D ratio, consistent with inhibitor-induced BRAF dimerization (*Figure 1b* and *Figure 1—figure supplement 3*). The change in A/D ratio with BRAF concentration observed at saturating inhibitor concentrations indicated that inhibitor-bound BRAF dimerizes with nanomolar affinity, representing several orders of magnitude enhancement over baseline dimerization.

We globally fit the FRET data to an allosteric model describing inhibitor-induced BRAF dimerization previously developed *Kholodenko, 2015*; *Figure 1c* and Materials and methods. In this model, dimerization is described by three equilibrium dissociation constants quantifying apo BRAF (B) dimerization ($K_D^{dimer}$: B+B⇌BB), dimerization with one drug/inhibitor (D) bound ($\frac{1}{2\alpha}K_D^{dimer}$: B+BD⇌BBD), and dimerization with two inhibitors bound ($\frac{1}{\alpha\beta}K_D^{dimer}$: BD+BD⇌BBDD). In turn, inhibitor binding is described by three dissociation constants quantifying binding to monomeric BRAF ($K_D^{drug}$: B+D⇌BD), binding to apo dimeric BRAF ($\frac{1}{2\alpha}K_D^{drug}$: BB+D⇌BBD), and binding to the second subunit of dimeric BRAF already harboring one inhibitor molecule ($\frac{2}{\beta}K_D^{drug}$: BBD+D⇌BBDD). To parameterize this model with experimental data, the FRET experiments were mapped onto the model using one fluorescence coefficient to describe the low-FRET monomeric forms of BRAF (B, BD) and a second coefficient to describe the high-FRET dimeric forms of BRAF (BB, BBD, BBDD) (see Materials and methods). Note that the factors of ½ and 2 in the above definitions arise from the stoichiometry of the reactions and the existence of two drug binding sites per dimer.

With the dimerization affinity of apo BRAF ($K_D^{dimer}$) constrained to a value of 62.4 μM (see above), global fitting produced well-constrained values for all other dissociation constants in the model (*Figure 1d and e* and *Figure 1—figure supplements 2b, c and 4*). Importantly, because of the explicit separation of inhibitor-driven dimerization into two steps, the energetic contributions of the first and second inhibitor molecules to dimerization can be resolved and are represented by the allosteric coupling parameters α and β, respectively. The parameter α can be interpreted as the degree to which baseline dimerization ($K_D^{dimer}$) is enhanced by a single bound inhibitor molecule ($\frac{1}{2\alpha}K_D^{dimer}$), while the parameter β quantifies any additional stabilization from a second inhibitor molecule ($\frac{1}{\alpha\beta}K_D^{dimer}$) (*Figure 1d*).

The αC-in RAF inhibitors can be divided into two classes based on binding mode. While both promote the αC-in state, type II inhibitors reach further into the active site than type I inhibitors and trigger an additional conformational change of the catalytic DFG-motif, in which the aspartate and phenylalanine DFG residues swap positions (*Figure 1—figure supplement 1c*; *Peng et al., 2015*; *Haling et al., 2014*). For both type I and type II inhibitors, our results showed that the total enhancement in BRAF dimerization from inhibitor binding, represented by the product αβ, is remarkably strong,

ranging from 3 to 5 orders of magnitude (*Figure 1f*). Surprisingly, for the type II inhibitors, including the clinical drugs ponatinib, belvarafenib, and tovorafenib, our global fitting analysis revealed that this allosteric coupling is not evenly distributed across the two drug binding sites of the dimer. Rather, these drugs are coupled to BRAF dimerization in a highly asymmetric manner, with the large majority of dimer promotion provided by the binding of the first inhibitor molecule (α values as large as $10^4$), and the contributions from the second inhibitor molecule being orders of magnitude smaller (β values no larger than $10^2$) (*Figure 1f and g* and *Figure 1—figure supplement 5*). In contrast, the allosteric coupling for the type I inhibitors is symmetrical (*Figure 1g*), with the α and β values being approximately equal (*Figure 1f* and *Figure 1—figure supplement 6*).

Because of the cyclic paths in the model, the pattern of allosteric coupling described above for dimerization affinities also applies to inhibitor affinities. Due to type II inhibitors having large α values and small β values, their binding affinity for apo BRAF dimers ($\frac{1}{2\alpha}K_D^{drug}$) is greatly enhanced compared to their binding affinity to BRAF monomers ($K_D^{drug}$), whereas binding to the second subunit of a partially occupied dimer ($\frac{2}{\beta}K_D^{drug}$) is only modestly enhanced (*Figure 1—figure supplement 5*). For many of the type II inhibitors the large α values boost affinities for the first dimer subunit into the picomolar range while the affinities for the second subunit are substantially weaker. In contrast, type I inhibitors, which have similar α and β values, bind the first and second subunits of the dimer with comparable affinity (*Figure 1e* and *Figure 1—figure supplement 6*).

In our FRET experiments inhibitor binding to BRAF monomers is not observed directly, and its weak affinity ($K_D^{drug}$) emerges as a prediction from the global fit analysis due to cyclic path constraints within the model. To support this prediction and confirm that our model is correctly parameterized, we performed control experiments with an intramolecular FRET sensor that directly detects inhibitor binding by measuring inhibitor-induced movements of the αC-helix (*Figure 1—figure supplement 7a and b*). Using a mutant construct (BRAF^DB) that cannot dimerize, we were able to uncouple inhibitor binding from BRAF dimerization and independently confirm the weak inhibitor affinity for BRAF monomers predicted by the allosteric model (see Materials and methods, *Figure 1—figure supplement 7c and d* ).

To independently verify that dimerization by type II inhibitors is driven predominately by α, we performed additional FRET experiments where the effects of β were eliminated by exploiting the inhibitor-blocking A481F mutation (*Hu et al., 2013*; *Figure 1—figure supplement 8a*). In control experiments with BRAF^A481F labeled with both donor and acceptor, inhibitor-induced dimerization was either not observed or was greatly weakened, confirming that the mutation effectively blocks inhibitor binding (*Figure 1—figure supplement 8b*). Acceptor-labeled BRAF^A481F and donor-labeled wild-type BRAF were mixed at equal molar ratios. Under these conditions, fully occupied wild-type BRAF dimers (BBDD) do not contribute to the FRET signal, and only the BRAF^A481F:BRAF heterodimers, which can bind only one inhibitor molecule (BBD) and are thus driven only by α, lead to observable FRET changes (*Figure 1—figure supplement 8a*). In these experiments, type II inhibitors triggered increases in the A/D ratios similar in magnitude to those observed in a donor-matched BRAF:BRAF experiment, indicating a similar extent of dimer induction (*Figure 1—figure supplement 9a*). Furthermore, dimerization affinities at saturating inhibitor concentrations were in the low nanomolar range, corresponding to a fold-increase over baseline dimerization of between $10^3$ and $10^4$, in good agreement with our measured α values for the type II inhibitors (*Figure 1—figure supplement 9b*). The type I inhibitor GDC0879, which possesses a far more modest α value (*Figure 1f*), failed to induce BRAF^A481F:BRAF heterodimers in this experiment. These results are consistent with the coupling between inhibitor binding and dimerization being highly asymmetric for type II inhibitors, where α is the dominant value, and more symmetric for type I inhibitors where α is smaller and similar in magnitude to β.

We also tested the effects of the αC-out inhibitors vemurafenib, dabrafenib, and encorafenib on BRAF dimerization. All three disrupted dimerization to such an extent that no dimerization signal was observed at saturating inhibitor concentrations, preventing the global fit analysis from converging to a constrained solution. To circumvent this, we used the oncogenic mutation E586K (BRAF^E586K) in the dimer interface to enhance dimerization (*Wan et al., 2004*), allowing us to obtain constrained α and β values for each αC-out inhibitor (*Figure 1f and h* and *Figure 1—figure supplement 10*). This analysis showed that dimer disruption by αC-out inhibitors is weaker than dimer promotion by αC-in inhibitors, with total decreases in dimerization affinity of 1–2 orders of magnitude (*Figure 1f* and *Figure 1—figure supplement 6*). Additionally, dimer disruption is not distributed equally between α and β, with

the former dominating for vemurafenib and the latter for dabrafenib and encorafenib (*Figure 1f*). These results demonstrate that our approach can quantify a wide range of inhibitor-induced dimerization effects including dimer promotion and disruption. Despite opposite effects on dimerization, both αC-in and αC-out RAF inhibitors can exhibit asymmetric allosteric coupling, suggesting that they are influenced by an asymmetry that is intrinsic to the BRAF kinase domain.

## Allosteric asymmetry drives accumulation of partially occupied BRAF dimers

To understand the functional consequences of asymmetric inhibitor-induced dimerization, we used our parameterized allosteric models to simulate the abundance of each BRAF biochemical species in solution (B, BD, BB, BBD, BBDD) as a function of inhibitor concentration (see Materials and methods). For all αC-in inhibitors, simulations predict a bell-shaped curve for the induction of partially occupied BBD dimers that increases with inhibitor concentration and peaks at approximately a 1:2 molar ratio of inhibitor to BRAF, before decreasing at higher concentrations due to the formation of dimers saturated with inhibitor (BBDD) (*Figure 2a* and *Figure 2—figure supplement 1*). The BBD induction amplitudes varied for type II inhibitors from 11% to 78% of total BRAF protein, but as a group were substantially higher than the amplitudes associated with the type I inhibitors which ranged from 4% to 11% (*Figure 2—figure supplement 2a*).

To determine how induction of BBD dimers is controlled by the α and β parameters, we used our allosteric model to simulate BBD formation over a wide range of α and β parameter space, while keeping apo dimerization affinity ($K_D^{dimer}$) and inhibitor affinity ($K_D^{drug}$) constant. The resulting BBD induction landscape is shown in *Figure 2b*. This analysis revealed that the amplitude of BBD induction is primarily dictated by the ratio between α and β, or the degree of asymmetry in the allosteric model, following a hyperbolic relationship with respect to α/β (*Figure 2c*), rather than by the total magnitude of dimer enhancement (αβ). This hyperbolic relationship was also observed with the experimentally parameterized models for the type I and type II inhibitors (*Figure 2d*). In fact, different α/β ratios fully account both for the differences between type I and type II inhibitors and for the variability within the type II class. For instance, among the clinically relevant type II drugs in our set, belvarafenib and ponatinib have the largest α/β ratios (>10³) and induce BBD dimers strongly, tovorafenib has a moderate α/β ratio (10²) and induces dimers moderately well, and sorafenib has the lowest α/β ratio (30) and is only marginally superior to the type I inhibitors at inducing BBD dimers (*Figure 2d*). These observations establish that greater allosteric coupling asymmetry translates into greater induction of partially occupied BBD dimers.

Comparing the inhibitors in our dataset in terms of their respective α and β values on the simulated BBD induction landscape clarifies that they occupy three distinct regions. Notably, the type I and type II inhibitors are resolved into separate groups (*Figure 2b*). The type II inhibitors are distributed in this space along an axis of increasing α/β ratio and approximately at right angles to an axis of increasing αβ, further highlighting the central role of allosteric asymmetry in determining the amplitude of BBD dimer induction. The αC-out inhibitors occupy a region of the landscape where both α and β are unfavorable, corresponding to dimer disruption rather than dimer promotion. This underscores that paradoxical activation by αC-out inhibitors is not driven by the dimer-induction mechanism of αC-in inhibitors, but by alternative mechanisms including negative allostery, RAS priming, and transactivation (*Jin et al., 2017*).

## Induction of partially occupied dimers quantitatively accounts for paradoxical activation in vitro

To test how the induction of BBD dimers translates into BRAF kinase activity, we used a fluorescence-based kinase assay to directly measure the phosphorylation of recombinant MEK by BRAF. In this assay, type II inhibitors induced strong dose-dependent increases in BRAF activity up to 19-fold above the no-inhibitor control that agreed strikingly well with simulated BBD induction curves (*Figure 2a* and *Figure 2—figure supplement 1 and 2b* ). In contrast, the type I inhibitors induced only relatively minor increases in BRAF activity (*Figure 2a* and *Figure 2—figure supplement 1 and 2b* ), consistent with their weak ability to induce BBD dimers, and the αC-out inhibitors vemurafenib and dabrafenib failed to trigger any discernable activation (*Figure 2—figure supplement 1*).

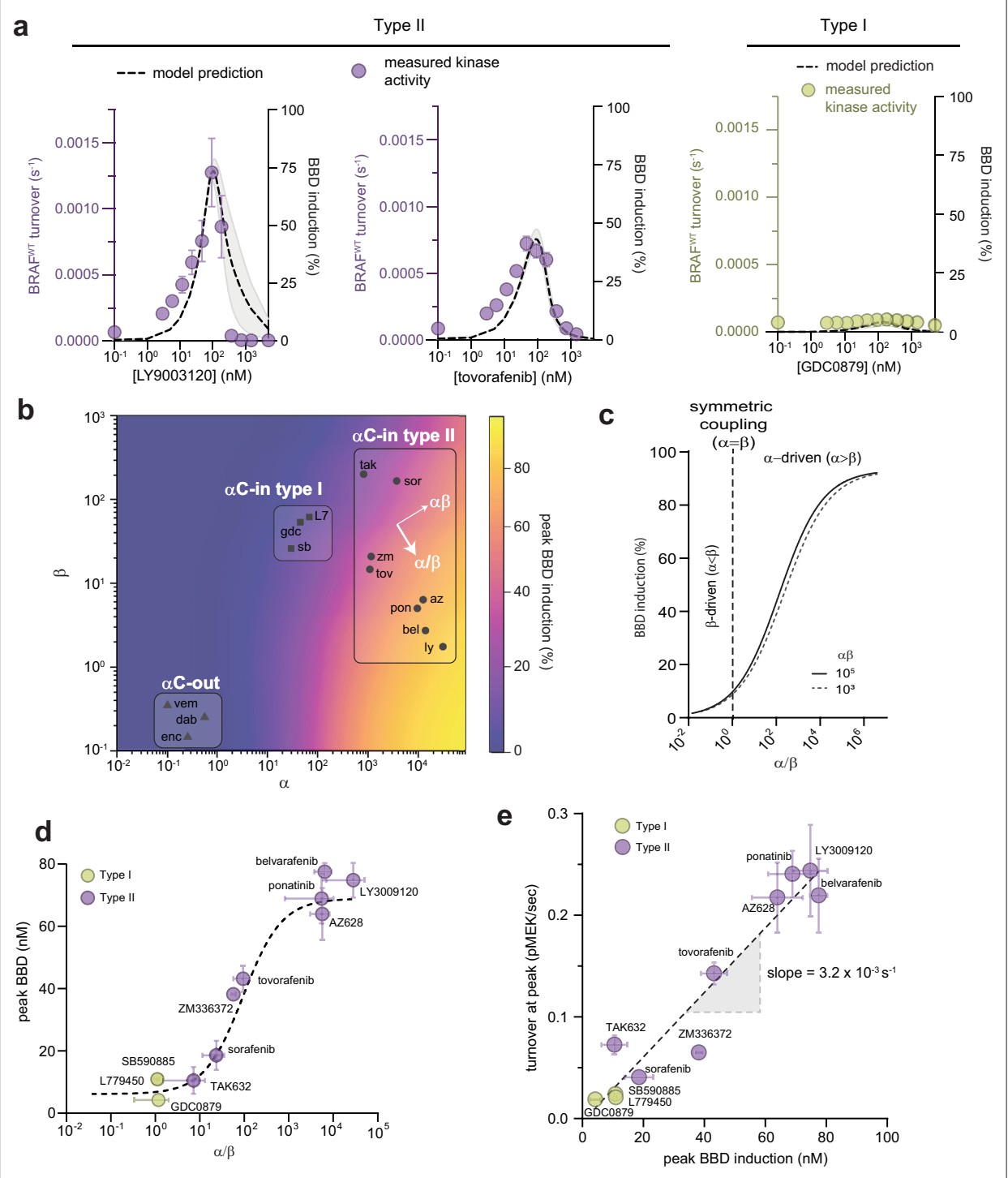

**Figure 2.** Allosteric asymmetry is the driving force for paradoxical activation by type II αC-in inhibitors. (**a**) Representative BRAF kinase activity data (circles, left y-axis) and induction of partially occupied BBD dimers (dashed line, right y-axis), for type II inhibitors LY3009120 and tovorafenib (purple) and the type I inhibitor GDC0879 (yellow). Activity data represent mean values ± s.e.m.; n=3 independent experiments each performed in duplicate. Activity data for other inhibitors are shown in *Figure 2—figure supplement 1*. BBD induction curves were simulated from the allosteric models parameterized with FRET data. The thickness of the band represents the 95%CI of the best-fit model from n=3 independently parameterized models. Simulations for other inhibitors are shown in *Figure 2—figure supplement 1*. (**b**) Induction landscape where the predicted amplitude of BBD induction is plotted over a wide range of α and β values. Simulations were performed using an allosteric model where $K_D^{dimer}$ and $K_D^{drug}$ were kept constant and α and β were systematically varied. Inhibitors are shown mapped onto the landscape (black symbols) based on their experimentally determined α and β factors. az, AZ628, bel, belvarafenib, dab, dabrafenib, enc, encorafenib, gdc, GDC0879, ly, LY30019120, L7, L779450, pon, ponatinib, sor, sorafenib, sb, SB590885,

*Figure 2 continued on next page*

*Figure 2 continued*

tov, tovorafenib, tak, TAK632, vem, vemurafenib, zm, ZM33637. (**c**) The simulated peak BBD induction is shown as a function of coupling asymmetry α/β at two fixed values of the total coupling strength αβ. (**d**) Simulated BBD induction magnitudes versus allosteric coupling ratios (α/β) for αC-in type I (yellow) and αC-in type II (purple) inhibitors. Data represent the mean ± s.e.m.; n≥3 independent experiments each performed in duplicate. The dashed line represents a hyperbolic fit to the data and confirms the relationship predicted in panel c. (**e**) Amplitude of BRAF kinase activation measured in vitro as a function of the simulated peak BBD induction for each inhibitor. Kinase activity data represent the mean ± s.e.m.; n=3 independent experiments each performed in duplicate. The slope of the linear fit, corresponding to the catalytic activity of BBD dimers, is indicated.

The online version of this article includes the following source data and figure supplement(s) for figure 2:

**Source data 1.** Tabulated source data for *Figure 2*.

**Figure supplement 1.** Simulations of partially occupied BRAF dimer formation and in vitro kinase activity.

**Figure supplement 1—source data 1.** Tabulated source data.

**Figure supplement 2.** BBD induction amplitude and peak catalytic turnover.

**Figure supplement 2—source data 1.** Tabulated source data.

Comparing the measured amplitudes of kinase activation with the BBD induction amplitudes predicted by the parameterized allosteric models for each inhibitor revealed an impressive linear correlation ($R^2$=0.93) (*Figure 2e*). The slope of the linear fit yielded a value for the catalytic turnover per BBD dimer of (3.2±0.3) × $10^{-3}$ s$^{-1}$. This value is in excellent agreement with the turnover value of 4.5×$10^{-3}$ s$^{-1}$ for drug-free BRAF dimers bound to 14-3-3 reported by *Liau et al., 2020*. This observation shows that partially occupied BRAF dimers are highly active and minimally impacted by the presence of one αC-in inhibitor molecule, and that the amplitude of paradoxical activation is directly determined by the concentration of this dimer species. The close correspondence between these independent experiments highlights the capability of our allosteric models to accurately predict how RAF inhibitors with a wide range of allosteric effects modulate the concentrations of different BRAF biochemical species to drive paradoxical activation. We conclude that paradoxical activation of BRAF homodimers by type II inhibitors occurs due to asymmetric allosteric coupling that selectively induces catalytically active BBD dimers, rather than fully inhibited BBDD dimers. Because the type I inhibitors lack allosteric asymmetry they cannot activate BRAF by this mechanism (*Figure 2d and e*).

We reasoned that the distinct allosteric coupling of type I and type II inhibitors may be due to the induction of different kinase conformations. Although both type I and II inhibitors are thought to promote the canonical αC-in state, the type II inhibitor ponatinib has been proposed to stabilize an intermediate 'αC-center' conformation (*Cotto-Rios et al., 2020*). Analysis of X-ray structures of BRAF in complex with several other type II inhibitors in our dataset, including AZ628 (*Karoulia et al., 2016*), LY3009120 (*Peng et al., 2015*), and TAK632 (*Okaniwa et al., 2013*), also suggested induction of intermediate αC-helix conformations, although crystal packing interactions might contribute to the differences (*Figure 3a* and *Figure 3—figure supplement 1a*). To test whether type I and type II inhibitors induce distinct αC-helix conformations in solution, we performed double electron-electron resonance (DEER) spectroscopy on BRAF by incorporating one nitroxide spin label onto the αC-helix (Q493C) and one onto the αG-helix (Q664C). This labeling arrangement yields shorter spin-spin distances for the αC-in state and longer distances for the αC-out state (*Figure 3b* and *Figure 3—figure supplement 1b and c*). Distance distributions derived from fitting of the DEER data (see Materials and methods) confirmed that all αC-in inhibitors tested induced shorter spin-spin distances relative to apo BRAF, consistent with promotion of the αC-in state. However, the type I inhibitor GDC0879 induced a shorter average spin-spin distance than the type II inhibitors AZ628, LY3009120, and TAK632, which yielded distance distributions that were intermediate between those observed with GDC0879 and with apo BRAF (*Figure 3b* and *Figure 3—figure supplement 1d*). These data confirm that the different binding modes of type I and type II inhibitors are associated with distinct αC-helix conformations. Since the conformation of the αC-helix is coupled to the N-lobe-to-C-lobe orientation of the kinase, which has been shown to modulate BRAF dimerization (*Lavoie et al., 2013*), these differences likely contribute to the distinct allosteric coupling patterns of type I and type II inhibitors.

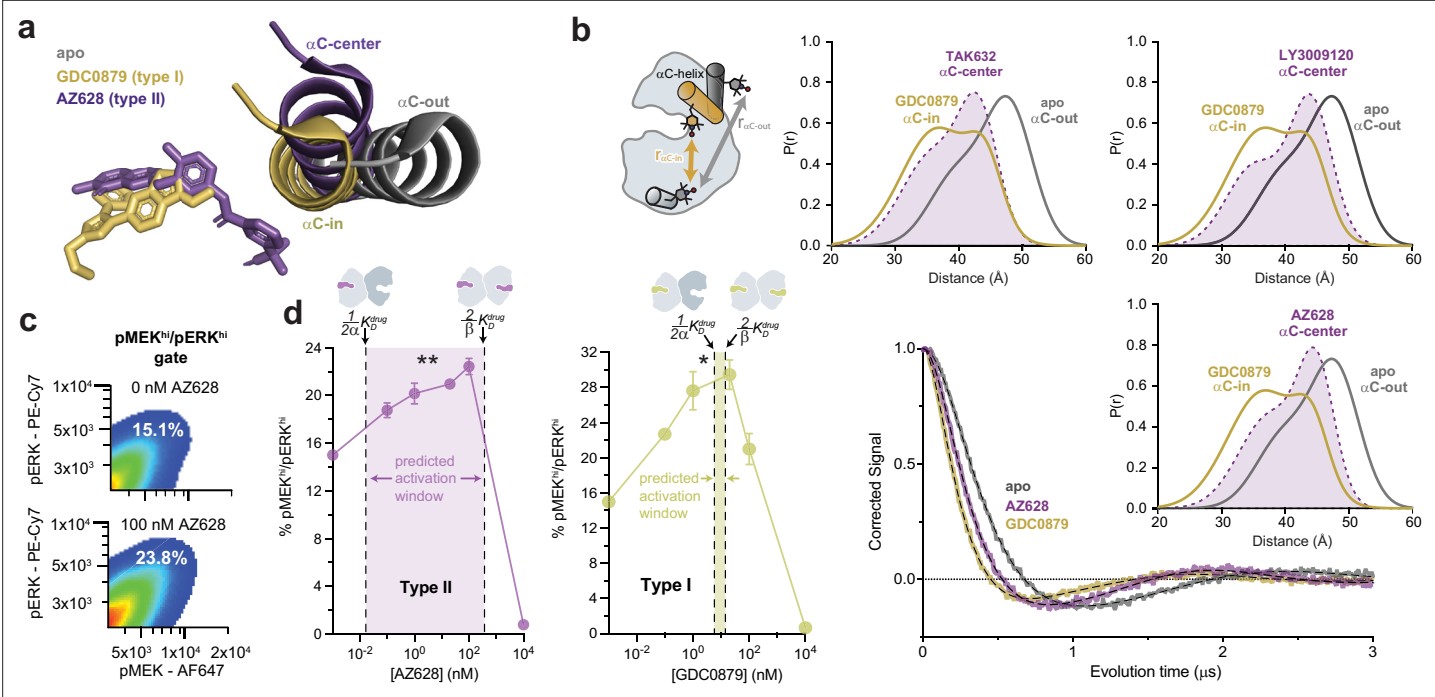

**Figure 3.** Type I and type II inhibitors induce distinct αC-helix conformations and promote MAPK/ERK pathway activation in SK-MEL-2 cells. (**a**) X-ray structures of BRAF in the apo state (*Park et al., 2019*), bound to the type I inhibitor GDC0879 (*Haling et al., 2014*), and bound to the type II inhibitor AZ628 (*Karoulia et al., 2016*) (PDB IDs: 6PP9, 4MNF, 4RZW), highlighting the different αC-helix conformations stabilized by each inhibitor. Structures were aligned on the C-terminal lobe. (**b**) (Left) Schematic of the labeling strategy used to track the conformation of the αC-helix with double electron-electron resonance (DEER). (Right) DEER waveforms and Gaussian distance distributions for apo BRAF (gray), BRAF bound to type I inhibitor GDC0879 (yellow), and BRAF bound to the type II inhibitors TAK632, AZ628, and LY3009120 (purple). (**c**) Representative flow-cytometry plots showing gating used to define maximally activated (pMEK$^{hi}$/pERK$^{hi}$) SK-MEL-2 cells in the absence (top) and presence (bottom) of AZ628 at an activating concentration of 100 nM. White labels indicate the % of live, single cells within the maximally activated gate. (**d**) Induction and inhibition of RAF-mediated MAPK phosphorylation in SK-MEL-2 cells treated with the type II inhibitor AZ628 (left, purple) and the type I inhibitor GDC0879 (right, yellow). The overlaid dashed lines represent the inhibitor affinities for the first and second binding sites on the dimer, defined by $\frac{1}{2\alpha}K_D^{drug}$ and $\frac{2}{\beta}K_D^{drug}$, respectively, determined from our global fitting analysis. The shaded concentration range between them defines the predicted activation window for each inhibitor. Data represent the mean ± s.e.m.; n=3 independent experiments. Significance was assessed using one-way ANOVA with Geisser-Greenhouse correction and Tukey's multiple comparison test. Asterisks reflect significant paradoxical activation of AZ pairs: [0→20 nM**p=0.0036], [0→100*p=0.0256] and GDC pairs: [0→0.1*p=0.0105], [0→20*p=0.0459]. For both inhibitor titrations, all pairwise comparisons [x→10,000 nM] also rose to significance and comparisons not mentioned did not. Induction and inhibition of RAF-mediated MAPK phosphorylation in SK-MEL-2 cells treated with additional inhibitors is shown in *Figure 3—figure supplement 2*.

The online version of this article includes the following source data and figure supplement(s) for figure 3:

**Source data 1.** Tabulated source data for *Figure 3*.

**Figure supplement 1.** Measuring the conformation of the αC-helix with double electron-electron resonance (DEER).

**Figure supplement 1—source data 1.** Tabulated source data.

**Figure supplement 2.** Inhibitor-induced MAPK/ERK pathway activation in SK-MEL-2 cells.

**Figure supplement 2—source data 1.** Tabulated source data.

**Figure supplement 3.** Western blot confirming the presence of all RAF isoforms in SK-MEL-2 cells.

**Figure supplement 3—source data 1.** Unedited western blots.

## Allosteric asymmetry creates a concentration window for paradoxical activation in cells

A key feature of our allosteric coupling models is that the asymmetry, represented by the α/β values, generates differences in the affinities for the first ($\frac{1}{2\alpha}K_D^{drug}$) and second inhibitor ($\frac{2}{\beta}K_D^{drug}$) molecules. We hypothesized that these differences would dictate the inhibitor concentration ranges at which paradoxical activation occurs in cells. To test this prediction, we assessed RAF-induced MAPK pathway

activation in SK-MEL-2 melanoma cells expressing wild-type BRAF and a gain-of-function variant of N-RAS (Q61R). Inhibitor-induced activation of MAPK signaling by RAF kinases was measured by flow cytometry, using intracellular staining with antibodies specific for phosphorylated (p)MEK1/2 and pERK1/2 (*Figure 3c*).

The type II inhibitors in our dataset all induced dose-dependent increases of pMEK1/2 and pERK1/2 indicating activation of the MAPK signaling pathway (*Figure 3d* and *Figure 3—figure supplement 2*; *Poulikakos et al., 2010*; *Karoulia et al., 2016*; *Yen et al., 2021*; *Cotto-Rios et al., 2020*; *Lavoie et al., 2013*). Several predictions from our in vitro allosteric model of paradoxical activation by αC-in RAF inhibitors are indeed reinforced by these experiments. First, many type II inhibitors activate the MAPK/ERK signaling pathway at inhibitor concentrations below 1 nM, consistent with ultrapotent inhibitor binding to the first site on the dimer that arises from the extreme allosteric coupling between inhibitor and BRAF dimerization (*Figure 3d* and *Figure 3—figure supplement 2* ). Second, the concentrations at which activation and inhibition occur differ by several orders of magnitude. These broad activation windows are consistent with the allosteric model, which predicts that the most asymmetric type II inhibitors including LY3009120, AZ628, ponatinib, and belvarafenib will have the largest activation windows (*Figure 3d* left panel and *Figure 3—figure supplement 2*). Indeed, plotting $\frac{1}{2\alpha}K_D^{drug}$ and $\frac{2}{\beta}K_D^{drug}$ onto the activation curves shows that the allosteric models provide reasonable bounds on the concentration regime in which paradoxical activation is observed in cells (*Figure 3d* and *Figure 3—figure supplement 2*).

Because of the symmetric allosteric coupling of the type I inhibitors the predicted affinities for each BRAF dimer subunit are comparable, indicating that there should be almost no activation window for these inhibitors. Nonetheless, we observed MAPK pathway activation by the type I inhibitors GDC0879, SB590885, and L779450 in SK-MEL-2 cells (*Figure 3d* right panel and *Figure 3—figure supplement 2*) across a wide concentration range, as seen with the type II inhibitors. This discrepancy suggests that MAPK activation by type I inhibitors is not mediated by BRAF homodimers but instead by BRAF:CRAF heterodimers. In support of this, activation by GDC0879 depends on the presence of CRAF (*Hatzivassiliou et al., 2010*), and both GDC0879 and SB590885 promote BRAF:CRAF heterodimers to a greater extent than the type II inhibitors AZ628 and sorafenib (*Hatzivassiliou et al., 2010*; *Jin et al., 2017*). Furthermore, type I inhibitors are less effective at inhibiting CRAF than type II inhibitors, preventing them from fully inhibiting the BRAF:CRAF heterodimers they induce (*Hatzivassiliou et al., 2010*; *Heidorn et al., 2010*). We have confirmed by immunoblot that SK-MEL-2 cells express all RAF isoforms at comparable levels, indicating that activation through BRAF:CRAF heterodimers is a plausible model for these inhibitors (*Figure 3—figure supplement 3*). These observations suggest that type I inhibitors activate by a distinct but related dimer-induction mechanism compared to the type II inhibitors, where differential drug binding to the two dimer subunits is achieved not by allosteric asymmetry but by the drugs having inherently different affinities for the BRAF and CRAF subunits.

## The BRAF dimer is not locked in the αC-in state but dynamically samples multiple conformations

To gain further insight into the mechanism underlying asymmetric allosteric coupling, we used $^{19}$F NMR to study the conformational dynamics of the BRAF αC-helix by incorporating a cysteine-reactive trifluoromethyl NMR probe 3-bromo-1,1,1-trifluoroacetone (BTFA) on the αC-helix (Q493C). Spectra of labeled apo BRAF showed two well-resolved resonances at –84.29 and –84.42 ppm (*Figure 4a*). Resonance assignment was achieved by adding saturating concentrations of ATP, known to induce the αC-out state. From this, the upfield resonance was defined as the αC-out (monomeric) state and the downfield resonance as the αC-in (dimeric) state (*Figure 4—figure supplement 1a*). Increasing the concentration of BRAF led to an increase in the αC-in peak area and a relative decrease in the αC-out peak area that fit to a monomer-dimer equilibrium with a $K_D$ of 32.2±9.4 μM, in reasonable agreement with the FRET experiments (*Figure 4—figure supplement 1b*).

Spectral deconvolution of the apo BRAF data revealed an additional resonance underneath the αC-in peak (*Figure 4a*). This second αC-in resonance, referred to as 'αC-in$^{broad}$', is substantially exchange-broadened, with a fitted line width approximately five times that of the narrower overlapping αC-in resonance, hereafter referred to as 'αC-in$^{narrow}$' (*Figure 4—figure supplement 1d*). The presence of the αC-in$^{broad}$ state was independently confirmed in transverse relaxation (T$_2$) experiments, where the peak intensity of the αC-in resonance was measured as a function of the transverse

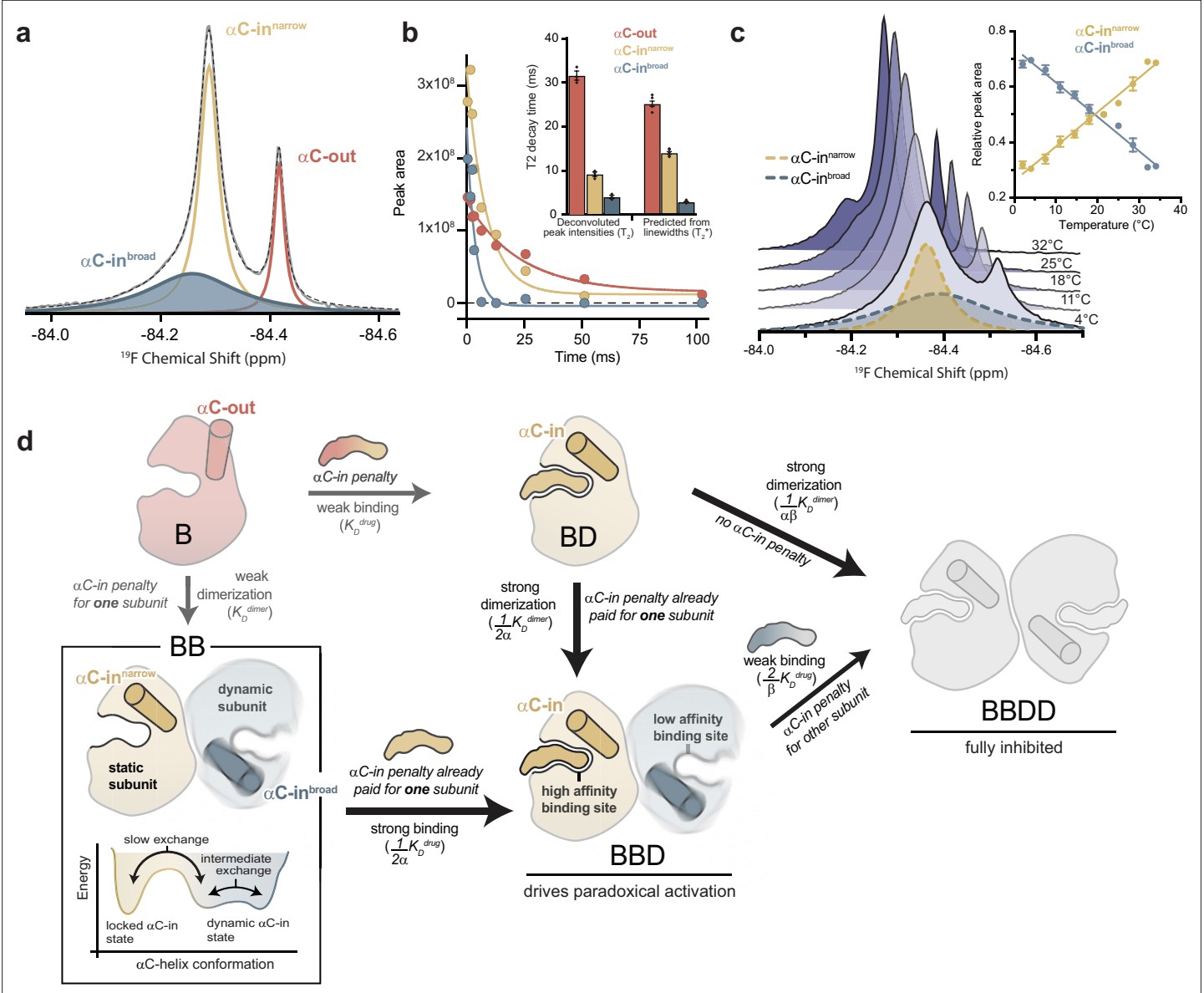

**Figure 4.** The αC-helix in the BRAF dimer dynamically samples multiple conformational states. (**a**) $^{19}F$ NMR spectrum of apo BRAF labeled on the αC-helix (Q493C) with 3-bromo-1,1,1-trifluoroacetone (BTFA). The raw spectrum (gray line) was fit to a multi-component Lorentzian model (dotted line). The deconvoluted spectrum consists of three unique resonances that correspond to one αC-out state (red) and two αC-in states (blue and yellow). Resonance assignments are shown in **Figure 4—figure supplement 1a,b and c** (**b**) Representative $^{19}F$ NMR $T_2$ relaxation profiles showing the peak areas of individual deconvoluted peaks shown in panel a as a function of $T_2$ decay time. $T_2$ decay parameters for each component were extracted by single-exponential fits. The inset shows a comparison of these $T_2$ values with $T_2^*$ values calculated from the spectral line widths of each component via the relationship $T_2^*=1/(\pi \times$ line width). Data represent the mean ± s.e.m.; n=4 independent experiments. (**c**) Variable-temperature $^{19}F$ NMR experiments. Spectra collected at lowest (light blue) to highest (dark blue) temperatures are shown. The deconvoluted dimer peaks are also shown as dotted lines for the lowest temperature. The inset shows the relative peak areas of αC-in$^{narrow}$ (yellow) and αC-in$^{broad}$ (blue) peaks as a function of temperature. Data represent the mean ± s.e.m.; n=3 independent experiments. (**d**) Schematic representation of our model for how dynamic heterogeneity in the BRAF dimer contributes to paradoxical activation. The relatively static αC-in$^{narrow}$ (yellow) and more dynamic αC-in$^{broad}$ (blue) states observed by $^{19}F$ NMR experiments are highlighted in the context of drug binding to monomers and dimers, with the biochemical species arranged in the same manner as in **Figure 1c**. This model can be further visualized with a free-energy diagram showing the αC-in$^{narrow}$ and αC-in$^{broad}$ states similarly populated and in slow exchange. The αC-in$^{broad}$ state consists of multiple conformational states separated by small energy barriers resulting in exchange on the intermediate timescale.

The online version of this article includes the following source data and figure supplement(s) for figure 4:

**Source data 1.** Tabulated source data for **Figure 4**.

*Figure 4 continued on next page*

*Figure 4 continued*

**Figure supplement 1.** $^{19}$F NMR resonance assignments and experiments confirming the presence of dynamic heterogeneity within the BRAF dimer.

**Figure supplement 1—source data 1.** Tabulated source data.

**Figure supplement 2.** Validation of BRAF labeling, stability, and purity.

**Figure supplement 2—source data 1.** Tabulated source data.

**Figure supplement 2—source data 2.** Raw gel from panel c.

---

magnetization evolution time and fit to an exponential decay model (*Figure 4—figure supplement 1e*). A double exponential fit was necessary to adequately describe these data (extra sum-of-squares F-test; p<0.0001), demonstrating the presence of two overlapping species with distinct relaxation times. Spectral deconvolution of these relaxation data produced an estimate of the relaxation times ($T_2$) for each peak that were in good agreement with calculated relaxation times ($T_2$*) derived from the observed line widths (*Figure 4b*). The short relaxation time of the αC-in$^{broad}$ resonance (<5 ms) is consistent with chemical exchange arising from the conformational dynamics of the αC-helix.

Four independent observations indicate that the αC-in$^{broad}$ resonance arises from a dimeric species: (1) the chemical shift of this species was similar to that of the αC-in$^{narrow}$ resonance (*Figure 4a*), (2) spectral deconvolution revealed a concentration dependence for the area of the αC-in$^{broad}$ peak that fits to a monomer-dimer equilibrium model closely agreeing with independent measurements (*Figure 4—figure supplement 1b*), (3) the αC-in$^{broad}$ resonance was not observed in BRAF samples with dimer-disrupting interface mutations (*Figure 4—figure supplement 1c*), (4) $D_2O$ exchange experiments revealed large and nearly identical isotope shifts for the αC-in$^{narrow}$ and αC-in$^{broad}$ resonances (0.047 vs 0.050 ppm), indicating a similar degree of solvent exposure of the probe, compared to a relatively small shift for the αC-out monomer peak (0.019 ppm) (*Figure 4—figure supplement 1f*).

Furthermore, as the experimental temperature was raised from 2°C to 34°C, the αC-in$^{broad}$ peak width decreased, consistent with faster exchange kinetics arising from increased conformational dynamics (*Figure 4c and d*). At the same time, the αC-in$^{broad}$ peak area decreased and the αC-in$^{narrow}$ peak area increased in a reciprocal manner, indicating a shifting equilibrium between the two dimeric states (*Figure 4c*). The equilibrium constant for this process, derived from the ratio of the integrated areas of the αC-in$^{broad}$ and αC-in$^{narrow}$ resonances, showed a linear dependence on temperature (*Figure 4—figure supplement 1g*).

Based on these results, we propose the following model (*Figure 4d*): The individual subunits of the BRAF dimer transition between two states with strikingly different dynamics. In one state, the subunit is locked in the αC-in conformation, with the αC-helix relatively static (αC-in$^{narrow}$). In the other, more dynamic state, the αC-helix undergoes transitions between conformational substates (αC-in$^{broad}$). Although our results do not define exactly how the two subunits of the dimer sample these different states, the dynamic αC-in state is populated under all experimental conditions. This indicates that BRAF dimers are not constrained in a symmetrical αC-in/αC-in configuration, as structural models suggest (*Park et al., 2019*; *Liau et al., 2020*), but rather that one subunit is free to explore other conformations. This explains how the binding of a single type II inhibitor molecule, which pays the energetic penalty for locking one BRAF subunit in the αC-in state, can dramatically increase BRAF dimerization, since the BRAF dimer only requires one and not both subunits to adopt the locked αC-in conformation. Equivalently, compared to the affinity for BRAF monomers, the drug affinity for dimers is greatly enhanced for one subunit and not for both because dimer formation only pays the energetic penalty of locking one subunit in the αC-in state. Thus, the dynamic heterogeneity of the BRAF dimer revealed by NMR may form the foundation for the asymmetric allosteric coupling that gives rise to paradoxical activation of BRAF by type II inhibitors.

## Discussion

The FDA-approved αC-out RAF inhibitors and the newer αC-in RAF inhibitors currently under development represent fundamentally different classes of drug that have opposite effects on RAF conformation and dimerization. Nonetheless, both drug classes trigger paradoxical activation of BRAF. Activation by the αC-out inhibitors is attributed in part to negative allostery, where dimer-disrupting allosteric effects prevent full occupancy of intact dimers (*Karoulia et al., 2016*). This model of negative

allostery does not provide an adequate explanation for paradoxical activation by αC-in inhibitors, which are positively coupled to dimerization and have been thought to bind both subunits of the dimer with equal potency (*Cotto-Rios et al., 2020*).

Here, we use a series of spectroscopic approaches paired with thermodynamic modeling and global fitting to quantify how inhibitor binding to each subunit of the BRAF dimer is coupled to BRAF dimerization. By explicitly separating inhibitor-induced dimerization into two steps, the allosteric effects from the binding of each inhibitor molecule, along with their respective binding affinities for each BRAF dimer subunit, are resolved. This analysis confirms that, unlike the dimer-disrupting αC-out inhibitors, the αC-in inhibitors dramatically increase BRAF dimerization affinity. Remarkably, for type II inhibitors this favorable allostery is not evenly distributed between both binding sites of the dimer, as previously proposed (*Cotto-Rios et al., 2020*), and instead occurs through an asymmetric coupling mechanism in which the binding of the first inhibitor molecule promotes dimerization much more strongly than binding of the second inhibitor molecule. This allosteric coupling also causes these inhibitors to have higher affinity for the first subunit of the dimer than for the second subunit. We demonstrate that this allosteric asymmetry triggers selective induction of BRAF dimers with only one subunit bound to inhibitor, with larger degrees of asymmetry leading to greater induction of these partially occupied dimers.

By combining our allosteric models of inhibitor-induced dimer induction with measurements of BRAF catalytic activity, we show that partially occupied BRAF dimers possess activity equivalent to fully activated inhibitor-free BRAF dimers, explaining their potent activating potential. Indeed, the degree of induction of partially occupied dimers fully accounts for both the concentration dependence and amplitude of BRAF activation measured in vitro for 11 different αC-in inhibitors. These results form the basis of a quantitative model for paradoxical activation of BRAF homodimers by αC-in inhibitors, where the apparent symmetry afforded by a dimer composed of identical subunits can be broken by asymmetric allosteric coupling between inhibitor binding and dimerization. The predictions of this model for the type II inhibitors are borne out in melanoma cells where the difference in affinities of the first and second inhibitor molecules for the dimer, $\frac{1}{2\alpha}K_D{}^{drug}$ and $\frac{2}{\beta}K_D{}^{drug}$, equivalent to the allosteric asymmetry α/β, is reflected in a wide concentration gap between where MAPK activation and inhibition are observed.

Comparing our mechanistic biophysical data with activation patterns observed in cells allowed us to uncover a key distinction between type I and type II inhibitors. While type II inhibitors are asymmetrically coupled and strongly activate BRAF homodimers, type I inhibitors lack sufficient allosteric coupling asymmetry to induce paradoxical activation through this mechanism. We demonstrate, however, that type I inhibitors induce MAPK pathway activation in melanoma cells, consistent with previous reports (*Poulikakos et al., 2010*; *Hatzivassiliou et al., 2010*; *Heidorn et al., 2010*). These observations point to type I inhibitors inducing paradoxical activation through BRAF:CRAF heterodimers, rather than BRAF homodimers, a hypothesis that is consistent with a large body of published work (*Hatzivassiliou et al., 2010*; *Karoulia et al., 2016*; *Jin et al., 2017*; *Heidorn et al., 2010*). Because type I inhibitors bind less tightly to CRAF than BRAF, the subunits of a heterodimer will have inherently different drug affinities, providing a window for paradoxical activation in the absence of allosteric asymmetry. Such a mechanism potentially applies to any dimer-promoting inhibitor with differing affinities for ARAF, BRAF, and CRAF, and would be expected to supplement asymmetry arising from the allosteric coupling mechanism we have uncovered. In particular, the emergence of clinical resistance to belvarafenib and another type II inhibitor, naporafenib, can be mediated by induction of ARAF (*Yen et al., 2021*; *Tkacik et al., 2023*). Since both belvarafenib and naporafenib bind ARAF less potently than BRAF, the disparity in drug affinities for the first and second subunits of ARAF/BRAF heterodimers is likely further boosted over that arising from asymmetric allosteric coupling alone, sustaining activation at higher drug concentrations and conferring resistance.

Exactly why type II inhibitors are more asymmetrically coupled to BRAF dimerization than type I inhibitors is not entirely clear. It is established that the conformational changes of the αC-helix induced by αC-in inhibitors also alter the relative orientations of the N-lobe and C-lobe of the kinase, improving the surface complementarity at the dimer interface and enhancing dimerization (*Lavoie et al., 2013*). We show that type I and type II inhibitors stabilize distinct αC-helix conformations in solution. In addition, type II inhibitors trigger a structural change of the catalytic DFG-motif and activation loop of BRAF that is not accomplished by type I inhibitors. It is likely that these collective differences unlock

the allosteric coupling asymmetry that defines the activating potential of type II inhibitors, but additional biophysical characterization is needed to confirm this.

Our data reveal that the subunits of the BRAF dimer are not symmetrically locked in the αC-in state, as suggested by static structural models, but rather exist in an equilibrium between a rigid αC-in state and a dynamic state in which the αC-helix conformation fluctuates on the μs-ms timescale. We interpret this as an indication that the BRAF dimer requires only one subunit to adopt the αC-in state, providing a straightforward explanation for why the binding of only one αC-in inhibitor molecule can dramatically increase dimerization affinity. The recent cryo-EM structure of dimeric BRAF in complex with the scaffolding protein 14-3-3, determined in the absence of inhibitors, shows an asymmetric arrangement of the BRAF dimer with respect to 14-3-3 (*Kondo et al., 2019*). This arrangement permits the C-terminal tail of one subunit to bind to and block the active site of the opposite subunit, while the reciprocal interaction is prevented. This further underscores the nonequivalence of the two subunits of the BRAF dimer and suggests that the dynamic asymmetry we observed in our NMR experiments may be an indication of broader functional asymmetry in BRAF signaling (*Hu et al., 2013*; *Lavoie et al., 2018*).

It is interesting to consider the preclinical and clinical experience with type II RAF inhibitors in light of our results. Several type II inhibitors studied here including belvarafenib (NCT04835805) and LY3009120 (NCT02014116) have entered clinical trials or could be repurposed like the FDA-approved chronic myeloid leukemia drug ponatinib. It is claimed that these molecules show minimal paradoxical activation compared to vemurafenib (*Yen et al., 2021*; *Cotto-Rios et al., 2020*). However, paradoxical activation by these type II inhibitors has been observed in cell lines both with and without elevated RAS activity (*Poulikakos et al., 2010*; *Karoulia et al., 2016*; *Yen et al., 2021*; *Cotto-Rios et al., 2020*; *Lai et al., 2022*; *Lavoie et al., 2013*), and demonstrated to trigger expression of ERK target genes and increased cell proliferation (*Lai et al., 2022*). We along with others show that ponatinib can activate ERK in RAS-mutant cells at concentrations as low as 1 nM, and that this activation is only suppressed at concentrations above 1 μM (*Cotto-Rios et al., 2020*), higher than the plasma level achievable in patients (*Ye et al., 2017*). Similarly, while belvarafenib has shown promising activity in preclinical and clinical studies (*Yen et al., 2021*), it increases MAPK signaling over a concentration range similar to other type II inhibitors like LY3009120, which has shown an unexpected lack of clinical efficacy (*Sullivan et al., 2020*) despite favorable preclinical results (*Peng et al., 2015*; *Yen et al., 2021*; *Chen et al., 2016*). Ultimately, whether paradoxical activation contributes to clinical outcomes for a particular inhibitor may depend on where the achievable dose lies with respect to the activation and inhibition sides of the paradoxical activation curves.

## Materials and methods
### Protein expression and purification
Human BRAF kinase domain (residues 448–723 with a tobacco etch virus [TEV]-protease-cleavable N-terminal 6x-His tag in pProEX) containing 16 mutations to improve solubility (*Tsai et al., 2008*) was expressed in chemically competent BL21 (DE3) RIL *Escherichia coli* (Agilent) for 18 hr at 18°C. Following sonication (Qsonica), lysates were clarified by centrifugation and loaded onto a HisTrap HP column (Cytiva). The column was washed with lysis buffer (50 mM Tris pH 8.0, 500 mM NaCl, 10% glycerol, 25 mM imidazole) and eluted with a 0–50% imidazole gradient over 12 column volumes using elution buffer (50 mM Tris pH 8.0, 500 mM NaCl, 10% glycerol, 500 mM imidazole). The His tag was cleaved overnight at 4°C with TEV protease. TEV protease was removed by an additional pass over a HisTrap HP column. Protein was further purified and desalted into desalt buffer (25 mM HEPES pH 7.5, 10% glycerol, 300 mM NaCl) using a Superdex S75 10/300 GL size exclusion column (Cytiva). Expression levels of BRAF mutants were increased by adding an N-terminal SUMO tag to the expression construct.

Human MEK1 kinase domain (residues 1–393 with a TEV-protease-cleavable N-terminal 6x-His tag in pET-Duet) containing an inactivating K97R mutation was expressed in chemically competent BL21 (DE3) RIL *E. coli* (Agilent) for 18 hr at 18°C. Cell pellets were sonicated in MEK lysis buffer (25 mM HEPES pH 7.5, 500 mM NaCl, 10% glycerol, 10 mM BME, 1 mM PMSF, 1× EDTA-free protease inhibitors [Roche], 20 mM imidazole) and clarified by centrifugation. Lysate was loaded onto a HisTrap HP column (Cytiva) and washed with MEK lysis buffer before eluting with a 0–50% imidazole gradient over

12 column volumes using elution buffer (25 mM HEPES pH 7.5, 500 mM NaCl, 10% glycerol, 10 mM BME, 300 mM imidazole). The His tag was cleaved overnight at 4°C with TEV protease. TEV protease was removed by an additional pass over the HisTrap HP column. Protein was further purified and desalted into desalt buffer (25 mM HEPES pH 7.5, 10% glycerol, 150 mM NaCl) using a Superdex S75 10/300 GL size exclusion column (Cytiva).

## FRET experiments tracking BRAF dimerization

We used the K547C site on the αD/αE-loop of BRAF to incorporate FRET dyes (*Figure 1—figure supplement 1a*). BRAF FRET samples were prepared by covalently labeling two separate pools of BRAF at 25 µM on K547C with either donor (Alexa Fluor 488 $C_5$ maleimide, Thermo Fisher) or acceptor (Alexa Fluor 568 $C_5$ maleimide, Thermo Fisher) at 0.8:1 molar ratio of fluorophore to protein, on ice. Labeling reactions were quenched at 1 hr with 1 mM DTT. Stopped flow fluorescence anisotropy experiments showed that the labeling kinetics of BRAF$^{K547C}$ were approximately 2 orders of magnitude faster than those of BRAF (*Figure 1—figure supplement 1b*), indicating that the K547C site can be selectively labeled without removing endogenous cysteines (C532, C685, C696). Mass spectrometry was used to confirm that only one cysteine was labeled with only one donor or acceptor fluorophore (*Figure 4—figure supplement 2a*). Donor-labeled and acceptor-labeled BRAF were then mixed at equal molar ratios and diluted into FRET buffer (25 mM HEPES pH 7.5, 300 mM NaCl, 10% glycerol, 10 mM MgCl$_2$, 1 mM EGTA, 2% DMSO). BRAF FRET sensor (49 µL) was then added to 384-well inhibitor titration plates containing 1 µL of inhibitor in DMSO prepared using a mosquito liquid handling robot (ttp Labtech) and incubated at room temperature for 90 min to ensure the reactions were at equilibrium. Dimerization experiments consisted of inhibitor titrations containing 12 concentrations including a DMSO-only control. Experiments for each inhibitor were done in duplicate at six different BRAF concentrations. Fluorescence data were recorded with a custom-built fluorescence plate reader (*Schaaf et al., 2017*) (Fluorescence Innovations) and contributions from the donor and acceptor fluorescence emission intensities (FRET A/D) were quantified by spectral unmixing using three basis functions for AF488 emission, AF568 emission, and Raman scattering (*Schaaf et al., 2017*).

## Global fitting analysis and thermodynamic modeling of FRET data

FRET A/D values were globally fit to a thermodynamic model describing inhibitor-induced BRAF dimerization (*Kholodenko, 2015*) shown in *Figure 1c* using the fitting and simulation software KinTek Explorer (https://kintekcorp.com/software; *Johnson et al., 2009*). In this model B and D represent monomeric BRAF and drug/inhibitor, respectively, and B, BB, BBD, BBDD represent apo monomeric BRAF, apo dimeric BRAF, dimeric BRAF partially occupied with one inhibitor molecule, and dimeric BRAF fully saturated with two inhibitor molecules. The equilibrium dissociation constants $K_D^{dimer}$ and $K_D^{drug}$ describe apo BRAF dimerization and inhibitor binding to monomeric BRAF, respectively. Microscopic reversibility restricts the specific values and relationships between each equilibrium constant within a cyclic path, such that the product of equilibrium constants along a cycle must equal one (*Kholodenko, 2015*; *Ederer and Gilles, 2007*). Consequently, each reaction within the model can be described in terms of either $K_D^{dimer}$ or $K_D^{drug}$ together with the allosteric coupling parameters α and β, which quantify the allosteric coupling in the system as described in the main text.

| Reaction | Equilibrium dissociation constant | Definition of thermodynamic factor |
|---|---|---|
| B+B⇌BB | $K_D^{dimer}$ | |
| B+D⇌BD | $K_D^{drug}$ | |
| BB +D⇌BBD | $K_3 = \frac{1}{2\alpha} \cdot K_D^{drug}$ | $2\alpha = \frac{K_D^{drug}}{K_3}$ |
| BD +B⇌BBD | $K_4 = \frac{1}{2\alpha} \cdot K_D^{dimer}$ | $2\alpha = \frac{K_D^{dimer}}{K_4}$ |
| BD +BD⇌BBDD | $K_5 = \frac{1}{\alpha\beta} \cdot K_D^{dimer}$ | $\beta/2 = \frac{K_4}{K_5}$ |
| BBD +D⇌BBDD | $K_6 = \frac{2}{\beta} \cdot K_D^{drug}$ | $\beta/2 = \frac{K_D^{drug}}{K_6}$ |

The factor of two in the above definitions arises from the stoichiometry of the reactions and the presence of two inhibitor binding sites within the BRAF dimer (*Kholodenko, 2015*).

The FRET A/D values were mapped onto the thermodynamic model using one fluorescence coefficient (C1) to describe monomeric and low-FRET forms of BRAF in solution (B, BD), and one fluorescence coefficient (C2) to describe dimeric high-FRET forms of BRAF in solution (BB, BBD, BBDD).

$$\frac{A}{D} = C1 \times \frac{(B + BD)}{(B + BD + 2(BB + BBD + BBDD))} + C2 \times \frac{2(BB + BBD + BBDD)}{(B + BD + 2(BB + BBD + BBDD))}$$

All experiments were fit to this model with the equilibrium dissociation constants $K_D^{dimer}$, $K_D^{drug}$, $\frac{1}{2\alpha}K_D^{dimer}$, $\frac{1}{\alpha\beta}K_D^{dimer}$, $\frac{1}{2\alpha}K_D^{drug}$, $\frac{2}{\beta}K_D^{drug}$ defined by $k_{-1}/k_1 = K_D$. The equilibrium dissociation constants were allowed to float in global fitting by locking the association rate constant to an arbitrarily high value beyond the diffusion limit ($k_1 = 1000$ nM$^{-1}$ s$^{-1}$) and allowing $k_{-1}$ to vary as described in the KinTek Explorer manual. The dimerization affinity of apo BRAF ($K_D^{dimer}$) was determined to be 62.4±2.9 μM in separate experiments using high concentrations of BRAF to maximize the dimerization signal (*Figure 1—figure supplement 2*). This value was then used to constrain $K_D^{dimer}$ in the fitting of other datasets, allowing lower concentrations of BRAF to be used. For BRAF concentrations above 2 μM an additional linear scale factor accounting for the secondary inner filter effect was required to fit the data. In the initial round of global fitting, the BRAF concentrations in the model, [BRAF]$_{1-6}$, were locked to their experimentally determined values. This fitting procedure yielded strongly asymmetric allosteric models for the type II inhibitors, and more symmetric models for the type I inhibitors. All parameters were constrained in these fits as determined by one-dimensional (1D) error surface analysis using chi2 thresholds calculated as previously described (*Figure 1—figure supplement 4a and b*; *Johnson et al., 2009*). Two-dimensional (2D) error surface analysis further confirmed that the pairs of equilibrium constants from which the α and β values are derived (e.g. $K_D^{dimer}$ and $\frac{1}{2\alpha}K_D^{dimer}$) are well constrained with respect to one another (*Figure 1—figure supplement 4c*). To investigate the effects of errors in protein concentration, a second round of global fitting was performed in which the [BRAF]$_{1-6}$ values were included as floating parameters in the fit, and 1D and 2D error surfaces used to confirm that the fits remained constrained despite the increased number of parameters (*Figure 1—figure supplements 2 and 4d*). For a subset of type II datasets, this procedure led to the appearance of double minima in the error surfaces, with one minimum corresponding to a highly asymmetric model and the other to a more symmetric model (*Figure 1—figure supplement 11*). To verify that the asymmetric models correspond to the correct solution, as indicated by the first round of fitting, inhibitor-induced BBD induction was simulated using the parameter values associated with both alternative models (*Figure 1—figure supplement 11b*) and compared to independently measured inhibitor-induced kinase activity (*Figure 1—figure supplement 11c*) to calculate a putative turnover number for the BBD dimer for each model solution (*Figure 1—figure supplement 11d*). Comparison with the catalytic turnover reported for 14-3-3-bound BRAF dimers (*Liau et al., 2020*; *Figure 1—figure supplement 11d*) showed that the asymmetric models were consistent with these published data (slightly lower activity than fully active BRAF dimers consistent with partial inhibition), whereas the symmetric models would require unrealistically high kinase activity for the BBD dimer (fivefold higher activity than fully active BRAF dimers despite partial inhibition). The results reported in the manuscript correspond to the values from the second round of global fitting.

## In vitro kinase activity assays

Kinase activity of BRAF K547C was measured using the FRET-based LanthaScreen kinase activity assay (Thermo Fisher). Kinase dead MEK1 K97R was labeled at 40 μM with Alexa Fluor 488 C$_5$ maleimide at a 1:1 molar ratio on ice for 1 hr. The labeling reaction was quenched using 1 mM DTT and desalted into 25 mM HEPES pH 7.5, 10% glycerol, and 150 mM NaCl. BRAF$^{15m}$ (BRAF$^{16m}$ with the additional E667F reversion mutation that restores MEK binding) was incubated at 400 nM with 2 μM MEK and 2× kinase buffer (50 mM HEPES pH 7.5, 0.2 mg/mL bovine γ-globulins, 20 mM MgCl$_2$, 600 mM NaCl, 2 mM EGTA) for 15 min. Inhibitor (1 μL) in 50% DMSO was then added to the BRAF/MEK reaction and incubated at room temperature for 1 hr. The kinase reaction was then initiated with the addition of 250 μM ATP for a final reaction concentration of 200 nM BRAF, 1 μM MEK, 100 μM ATP, 1× kinase buffer, and 5% DMSO with inhibitor and incubated for 60 min. Reactions were quenched with a 2× dilution into TR-FRET buffer (Thermo Fisher) with 40 mM EDTA and 4 nM LanthaScreen Tb-pMAP2K1 (pSer 217/

pSer 221) antibody (Thermo Fisher) and incubated at room temperature for 2 hr. The TR-FRET ratio was measured using a Tecan M1000 pro plate reader with excitation at 340 nm followed by a 100 µs delay before reading emission at 490 nm (donor) and 520 nm (acceptor) with a 200 µs integration time. Increases in kinase activity were inferred from increases in FRET (A/D ratio). Kinase turnover ($s^{-1}$) was interpolated from a phoshoMEK1 standard curve. Outliers were identified and removed from analysis using the ROUT method in GraphPad Prism with a recommended Q coefficient of 1%.

## Flow cytometry

SK-MEL-2 melanoma cells (purchased from ATCC and used directly for experiments) were seeded into 96-well V-bottom plates at $1\times10^7$ cells/mL in 100 µL Dulbecco's Modified Eagle Medium (Corning) supplemented with 100 U/mL penicillin-streptomycin and 10% FBS. Cells were rested for 1 hr and treated for 1 hr at 37°C with RAF inhibitor or DMSO (vehicle) to a final concentration of 0.1% DMSO. Cells were then fixed in 4% paraformaldehyde for 20 min at 4°C. Cells were then washed twice in FACS buffer (PBS, 2% FBS, 2 mM EDTA) and permeabilized in BD Phosflow Perm Buffer III (BD Biosciences), according to the manufacturer's instructions. Cells were then stained in FACS buffer with anti-pMEK1/2 (BD Biosciences) and anti-pERK1/2 (BD Biosciences). After washing, samples were analyzed on a BD Fortessa X-30 flow cytometer. Using FlowJo software (TreeStar), live, single cells were selected for further analysis (gated) via characteristic laser side scatter (SSC, 90°) vs. forward scatter (FSC, in-path) area and magnitude profiles. Gates quantifying the frequency of maximally signaling pMEK$^{hi}$/pERK$^{hi}$ cells within this population were placed using a quadrant with an arbitrary cutoff of 15% for each vehicle control. Gate positions were then copied to other samples for identical positioning across inhibitor concentrations within the same staining panel and experiment day. Data were analyzed in Prism (GraphPad), with significance assessed by one-way ANOVA with Tukey's correction for multiple comparisons, with pMEK$^{hi}$/pERK$^{hi}$ cell frequencies compared to all other inhibitor doses within each inhibitor titration.

## Immunoblotting

Immunoblotting was performed as previously described (*Brian et al., 2022*; *Brian et al., 2020*). Briefly, 0.025 million cell equivalents of whole cell lysate were run through 7% Tris-acetate polyacrylamide gels and transferred to polyvinylidene difluoride membranes. Membranes were blocked in Intercept (TBS) Blocking Buffer (LI-COR Biosciences) for 1 hr, and then incubated overnight with primary antibody: ARAF (Cell Signaling), BRAF (Cell Signaling), CRAF (Cell Signaling), MEK1/2 (Cell Signaling), and ERK1/2 (Cell Signaling). Membranes were washed and incubated for 1 hr at room temperature with corresponding species-reactive secondary antibody and then imaged using an Odyssey CLx near-infrared imager (LI-COR Biosciences).

## DEER spectroscopy

DEER samples were prepared by labeling 10 µM BRAF containing three dimer-breaking mutations (BRAF$^{DB}$,R509H, L515G, M517W) (*Röring et al., 2012*) on the αC-helix (Q493C) and the αG-helix (Q664C) with a twofold excess of 4-maleimido-TEMPO for 45 min at 4°C. Spin-labeled BRAF was concentrated to 60–80 µM, buffered in $D_2O$ with 25 mM HEPES pH 7.5, 500 mM NaCl, and 10% $d_8$-glycerol and rapidly frozen in 1.1 mm ID/1.6 mm OD quartz capillary tubes using liquid nitrogen-cooled isopropanol. For samples containing inhibitors, prior to freezing, BRAF was incubated for 90 min with a fivefold molar excess of inhibitor dissolved in deuterated DMSO. DEER spectra were collected at 65 K on an Elexsys E580 spectrometer (Bruker) equipped with an EN5107 resonator operating at Q-band frequencies using parameters previously described (*Majumdar et al., 2021*) Data were analyzed using custom software (https://github.com/thompsar/Venison, copy archived at *Thompson, 2022*) written in Python and based on DeerAnalysis 2017. DEER data were phased and background-corrected using a homogeneous background model to derive the DEER waveform. Distance distributions were obtained by fitting these waveforms using unconstrained Tikhonov regularization, with smoothing parameter $\lambda$ chosen using the L-curve method and leave-one-out cross-validation. Features of the DEER waveform that contributed to unstable populations that were distinct from the primary populations and beyond the sensitivity limit of the 6 µs evolution time (~60 Å) were suppressed by incorporation into the background model. The distribution obtained by Tikhonov regularization using this corrected waveform was used to initialize fitting of the waveforms to a sum of

Gaussians model that describes the centers of the spin-spin distances, as well as the widths and mole fractions. The number of subpopulations was determined by selecting the fewest number of Gaussian centers that met the RMSD minimization threshold calculated by the Bayesian information criterion. Distance distributions were in good agreement with distributions calculated from X-ray structures of the αC-in and αC-out states using MtsslSuite (http://www.mtsslsuite.isb.ukbonn.de/; *Hagelueken et al., 2012*).

## $^{19}$F NMR spectroscopy

BRAF $^{19}$F samples were prepared by covalently labeling 50 µM of BRAF$^{16m}$ on a single αC-helix cysteine (Q493C) with a 1.75 molar excess of BTFA for 1 hr at 4°C. Samples were quenched with 1 mM DTT and desalted into NMR buffer (25 mM HEPES pH 7.5, 500 mM NaCl, 10% glycerol) supplemented with 10% $D_2O$ and 0.005% trifluoroacetic acid as an internal reference. The presence of only one BTFA probe on the BRAF kinase domain was verified by mass spectrometry (*Figure 4—figure supplement 2a*). Circular dichroism temperature denaturation experiments were carried out to verify that the BTFA probe did not alter BRAF stability at the temperatures used in NMR experiments (*Figure 4—figure supplement 2b*). $^{19}$F NMR experiments were performed at 298 K using a Bruker 600 MHz Avance NEO equipped with a 5 mm cryogenic triple resonance probe tuned to 565.123 MHz. 1D spectra were collected using the zg pulse program (Bruker TopSpin 4.1.4) with a 13.5 µs 90° pulse time, 0.2 s acquisition time, and a 1 s D1 relaxation delay time. Variable-temperature experiments involved a 5 min sample equilibration period at each temperature prior to collecting spectra. Transverse relaxation ($T_2$) experiments were performed using the Carr-Purcell-Meiboom-Gill pulse sequence with a 12 µs 90° pulse time, 2 s D1 relaxation delay, 200 µs $D_2O$ fixed spin-echo time, and a 24 µs 180° refocusing period. Spectra were acquired with 2048 scans with total transverse magnetization times of 0.4, 1.6, 2.4, 3.2, 6.4, 12.8, 25.6, 51.2, and 102.4 ms. 1D spectra were processed using MestReNova 14.3.0 by aligning the TFA reference to –75.32 ppm, applying automatic zeroth- and first-order phase corrections, a 3° polynomial Bernstein baseline correction, and 1 Hz line broadening correction.

$T_2$ relaxation profiles were created by measuring the intensity at –84.42 ppm (αC-out) and –84.29 ppm (αC-in) as a function of delay time and fit to both single and double exponential decay models in GraphPad Prism 9.4.0 with the double exponential being the preferred model for both resonances as determined by an extra sum-of-squares F-test (p<0.0001). In a separate $T_2$ analysis, each spectrum was subjected to spectral deconvolution using OriginPro 2022 by fitting the 1D spectra to a three-component Lorentzian model. The time dependence of the component amplitudes is shown in *Figure 4b*.

## Mass spectrometry analysis of commercial kinase inhibitors

Commercially available RAF kinase inhibitors were purchased from Selleckchem and TargetMol. High-resolution mass spectrometry data were collected on a Bruker BioTOF II instrument with an infusion electrospray ionizer. Compounds were dissolved in DMSO to a concentration of 10 mM. Stocks were diluted 100× with MeOH and injected at a rate of 10 µL/min. Mass spectrometry was run and analyzed in positive-ion mode with either a PEG 600 or PEG 400 internal standard (see *Supplementary file 1*). Data were analyzed using Bruker Data Analysis Software.

## Code availability

The program used to analyze DEER data for this study, Venison, is available for download from https://github.com/thompsar/Venison, (copy archived at *Thompson, 2022*).

## Acknowledgements

We thank Frank Sicheri and Scott Prosser for insightful conversations, Frank Sicheri for the BRAF$^{16m}$ construct, and Donita Brady for the human MEK construct. We thank Todd Rappe for help with NMR experiments, Joseph Dalluge for help with mass spectrometry, and Jaclyn Frank for assistance with manuscript preparation. DMR acknowledges support from F31CA257218 and T32GM132029, JTG acknowledges support from American Cancer Society – Kirby Foundation Postdoctoral Fellowship PF-21-068-01-LIB, TSF acknowledges support from R01AR073966, NML acknowledges support from R33CA246363, and WCKP acknowledges support from 5R35GM140837.

## Additional information

### Funding

| Funder | Grant reference number | Author |
|---|---|---|
| National Cancer Institute | F31CA257218 | Damien M Rasmussen |
| National Cancer Institute | R33CA246363 | Nicholas M Levinson |
| National Institute of General Medical Sciences | T32GM132029 | Damien M Rasmussen |
| American Cancer Society – Kirby Foundation | PF-21-068-01-LIB | Joseph T Greene |
| National Institute of Arthritis and Musculoskeletal and Skin Diseases | R01AR073966 | Tanya S Freedman |
| National Institute of General Medical Sciences | 5R35GM140837 | William CK Pomerantz |

The funders had no role in study design, data collection and interpretation, or the decision to submit the work for publication.

### Author contributions

Damien M Rasmussen, Conceptualization, Data curation, Validation, Visualization, Methodology, Writing – original draft, Writing – review and editing, Formal analysis, Funding acquisition, Investigation; Manny M Semonis, Joseph T Greene, Investigation, Visualization, Methodology, Formal analysis; Joseph M Muretta, Conceptualization, Software, Validation; Andrew R Thompson, Software, Validation, Investigation, Methodology; Silvia Toledo Ramos, Investigation, Methodology; David D Thomas, Validation, Methodology; William CK Pomerantz, Validation, Investigation, Methodology; Tanya S Freedman, Validation, Investigation, Visualization, Methodology, Writing – review and editing; Nicholas M Levinson, Conceptualization, Resources, Data curation, Supervision, Funding acquisition, Validation, Investigation, Visualization, Methodology, Project administration, Writing – review and editing

### Author ORCIDs

Damien M Rasmussen http://orcid.org/0000-0001-6828-8520
David D Thomas http://orcid.org/0000-0002-8822-2040
Tanya S Freedman http://orcid.org/0000-0001-5168-5829
Nicholas M Levinson http://orcid.org/0000-0003-4338-8087

Reviewer #1 (Public Review): https://doi.org/10.7554/eLife.95481.2.sa1
Reviewer #2 (Public Review): https://doi.org/10.7554/eLife.95481.2.sa2
Author response https://doi.org/10.7554/eLife.95481.2.sa3

## Additional files

### Supplementary files

• Supplementary file 1. Inhibitor sources and validation by mass spectrometry.

### Data availability

All source data are available in this paper and supplementary information. The program used to analyze DEER data for this study, Venison, is available for download from https://github.com/thompsar/Venison, copy archived at *Thompson, 2022*.

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
