## [Editor Report · eLife assessment]

This elegant study presents **important** findings into how small molecules that were originally developed to inhibit the oncogenic kinase, BRAF, instead trigger activation of this kinase target. **Compelling** and comprehensive evidence supports a new allosteric model to explain the paradoxical activation. This rigorous work will be of great interest to biochemists, structural biologists, and those working on strategies to inhibit kinases in the context of human disease.

---

## [Referee Report · Reviewer #1 (Public Review)]

Summary:

The authors quantitatively describe the complex binding equilibria of BRAF and its inhibitors resulting in some cases in the paradoxical activation of BRAF dimer when bound to ATP competitive inhibitors. The authors use a biophysical tour de force involving FRET binding assays, NMR, kinase activity assays, and DEER spectroscopy.

Strengths:

The strengths of the study are the beautifully conducted assays that allow for a thorough characterization of the allostery in this complex system. Additionally, the use of F-NMR and DEER spectroscopy provides important insights into the details of the process.

The resulting model for binding of inhibitors and dimerization (Figure 4) is very helpful.

Weaknesses:

This is a complex system and its communication is inherently challenging. It might be of interest to the broader readership to understand the implications of the model for drug development and therapy.

---

## [Referee Report · Reviewer #2 (Public Review)]

Summary:

This manuscript uses FRET, 19F-NMR, and DEER/EPR solution measurements to examine the allosteric effects of a panel of BRAF inhibitors (BRAFi). These include first-generation aC-out BRAFi, and more recent Type I and Type II aC-in inhibitors. Intermolecular FRET measurements quantify Kd for BRAF dimerization and inhibitor binding to the first and second subunits. Distinct patterns are found between aC-in BRAFi, where Type I BRAFi binds equally well to the first and second subunits within dimeric BRAF. In contrast, Type II BRAFi shows stronger affinity for the first subunit and weaker affinity for the second subunit, an effect named "allosteric asymmetry". Allosteric asymmetry has the potential for Type II inhibitors to promote dimerization while favoring occupancy of only one subunit (BBD form), leading to the enrichment of an active dimer.

Measurements of in vitro BRAF kinase activity correlate amazingly well with the calculated amounts of the half-site-inhibited BBD forms with Type II inhibitors. This suggests that the allosteric asymmetry mechanism explains paradoxical activation by this class of inhibitors. DEER/EPR measurements further examine the positioning of helix aC. They show systematic outward movement of aC with Type II inhibitors, relative to the aC-in state with Type I inhibitors, and further show that helix aC adopts multiple states and is therefore dynamic in apo BRAF. This makes a strong case that negative cooperativity between sites in the BRAF dimer can account for paradoxical kinase activation by Type II inhibitors by creating a half-site-occupied homodimer, BBD. In contrast, Type I inhibitors and aC-out inhibitors do not fit this model, and are therefore proposed to be explained by previously proposed models involving negative allostery between subunits in BRAF-CRAF heterodimers, RAS priming, and transactivation.

Strengths:

This study integrates orthogonal spectroscopic and kinetic strategies to characterize BRAF dynamics and determine how it impacts inhibitor allostery. The unique combination of approaches presented in this study represents a road map for future work in the important area of protein kinase dynamics. The work represents a worthy contribution not only to the field of BRAF regulation but to protein kinases in general.

Weaknesses:

Some questions remain regarding the proposed model for Type II inhibitors and its comparison to Type I and aC-out inhibitors that would be useful to clarify. Specifically, it would be helpful to address whether the activation of BRAF by Type II inhibitors, while strongly correlated with BBD model predictions in vitro, also depends on CRAF via BRAF-CRAF in cells and therefore overlaps with the mechanisms of paradoxical activation by Type I and aC-out inhibitors.

---

## [Author Response]

We would like to thank the reviewers for their helpful comments. We note that both reviews are strongly supportive with comments including, “a biophysical tour de force” (rev #1), “the study is exemplary” (rev #2), and “represents a roadmap for future work” (rev #2). Below we respond to each reviewer comment.

**Reviewer #1**
This study provides a detailed and quantitative description of the allosteric mechanisms resulting in the paradoxical activation of BRAF kinase dimers by certain kinase inhibitors. The findings provide a much needed quantiative basis for this phenomenon and may lay the foundation for future drug development efforts aimed at the important cancer target BRAF. The study builds on very evidence obtained by multiple independent biophysical methods.Summary:The authors quantitatively describe the complex binding equilibria of BRAF and its inhibitors resulting in some cases in the paradoxical activation of BRAF dimer when bound to ATP competitive inhibitors. The authors use a biophysical tour de force involving FRET binding assays, NMR, kinase activity assays and DEER spectroscopy.

We are gratified by the reviewer’s supportive summary.

Strengths:The strengths of the study are the beautifully conducted assays that allow for a thorough characterization of the allostery in this complex system. Additionally, the use of F-NMR and DEER spectroscopy provide important insights into the details of the process.The resulting model for binding of inhibitors and dimerization (Fig.4) is very helpful.Weaknesses:This is a complex system and its communication is inherently challenging. It might be of interest to the broader readership to understand the implications of the model for drug development and therapy.

We agree with the reviewer that this is a complicated system. With regard to inhibitor development, a key insight is that designing aC-in state inhibitors that avoid paradoxical activation may be non-trivial because these molecules not only induce dimers but also tend to bind the second dimer subunit more weakly than the first, due to allosteric asymmetry and/or inherently different affinities for each RAF isoform. We feel the full implications for future therapeutic development are an extensive topic that is beyond the scope of our work, which is focused on the properties of current inhibitors.

Recommendations for the author:The experimental work, analysis and resulting model are excellent. I had some difficulty following the complex model in some instances and it may be useful to review the description of the model and see whether it can be made more palatable to the broader readership.I think it would be useful to discuss the model presented in reference 40 (Kholodenko) and to compare it to the presented model here.

We regret any confusion with regards to the nature of the model. Our analysis was built upon the model developed by Boris Kholodenko as reported in his 2015 Cell Reports paper. This formed the theoretical framework that combined with our experimental data allowed us to parameterize this model to obtain experimental values for the equilibrium constants and allosteric coupling factors.

**Reviewer #2**
This manuscript combines elegant biophysical solution measurements to address paradoxical kinase activation by Type II BRAF inhibitors. The novel findings challenge prevailing models, through experiments that are rigorous and carefully controlled. The study is exemplary in the breadth of strategies it uses to address protein kinase dynamics and inhibitor allostery.Summary:This manuscript uses FRET, 19F-NMR and DEER/EPR solution measurements to examine the allosteric effects of a panel of BRAF inhibitors (BRAFi). These include first-generation aC-out BRAFi, and more recent Type I and Type II aC-in inhibitors. Intermolecular FRET measurements quantify Kd for BRAF dimerization and inhibitor binding to the first and second subunits. Distinct patterns are found between aC-in BRAFi, where Type I BRAFi bind equally well to the first and second subunits within dimeric BRAF. In contrast, Type II BRAFi show stronger affinity for the first subunit and weaker affinity for the second subunit, an effect named "allosteric asymmetry". Allosteric asymmetry has the potential for Type II inhibitors to promote dimerization while favoring occupancy of only one subunit (BBD form), leading to enrichment of an active dimer.Measurements of in vitro BRAF kinase activity correlate amazingly well with the calculated amounts of the half site-inhibited BBD forms with Type II inhibitors. This suggests that the allosteric asymmetry mechanism explains paradoxical activation by this class of inhibitors. DEER/EPR measurements further examine the positioning of helix aC. They show systematic outward movement of aC with Type II inhibitors, relative to the aC-in state with Type I inhibitors, and further show that helix aC adopts multiple states and is therefore dynamic in apo BRAF. This makes a strong case that negative cooperativity between sites in the BRAF dimer can account for paradoxical kinase activation by Type II inhibitors by creating a half site-occupied homodimer, BBD. In contrast, Type I inhibitors and aC-out inhibitors do not fit this model, and are therefore proposed to be explained by previous proposed models involving negative allostery between subunits in BRAF-CRAF heterodimers, RAS priming, and transactivation.Strengths:This study integrates orthogonal spectroscopic and kinetic strategies to characterize BRAF dynamics and determine how it impacts inhibitor allostery. The unique combination of approaches presented in this study represents a road map for future work in the important area of protein kinase dynamics. The work represents a worthy contribution not only to the field of BRAF regulation but protein kinases in general.Weaknesses:Some questions remain regarding the proposed model for Type II inhibitors and its comparison to Type I and aC-out inhibitors that would be useful to clarify. Specifically, it would be helpful to address whether the activation of BRAF by Type II inhibitors, while strongly correlated with BBD model predictions in vitro, also depends on CRAF via BRAF-CRAF in cells and therefore overlaps with the mechanisms of paradoxical activation by Type I and aC-out inhibitors.

We agree with the reviewer that this is a worthy question to be pursued. However, given the substantial experimental effort required for such an endeavor, and the highly supportive nature of the reviewer comments, including that “This is a strong manuscript that I feel is well above the bar for publication”, we believe this effort is more appropriate for a future study.

This is a strong manuscript that I feel is well above the bar for publication. Nevertheless, it is recommended that the authors consider addressing the following points in order to support their major conclusions.(1) Fig 3D shows similar effects of Type II and Type I inhibitors in the biphasic increase of cellular pMEK/pERK. From this, the authors argue that Type II inhibitors are explained by negative allostery in the BRAF homodimer (based on Fig 2E), while Type I inhibitors are not. But it seems possible that despite the terrific correlation between BBD and BRAF kinase activities measured in vitro, CRAF is still important to explain pathway activation in cells. It also seems conceivable that the calculated %BBD between different Type II inhibitors may not correlate as well with their effects on pathway activation in cells. These possibilities should be addressed.

We agree with the reviewer that it is likely that CRAF contributes to paradoxical activation by type II inhibitors in cells. It is also likely that other cellular factors such as RAS-priming and membrane recruitment play a role in activation. However, we note that for the type II inhibitors there is good agreement between the biophysical predictions and the concentration regimes in which activation is observed in cells, suggesting that these predictions are capturing a key part of the activation process that occurs in cells.

(2) In Fig 2A, is it possible to report the activity of dimeric BRAF-WT in the absence of inhibitor? This would help confirm that the maximal activity measured after titrating inhibitor is indeed consistent with the predicted %BBD population, which would be expected to have half of the specific activity of BB.

In principle, it is possible to determine the catalytic activity of apo dimers (BB) by combining our model predictions for the concentration of BB dimers and our activity measurements. However, because the activity assays are performed at nanomolar kinase concentrations, whereas the baseline dimerization affinity of BRAF is in the micromolar range, the observed activity of apo BRAF arises from a small subpopulation of dimers (on the order of 4 percent under the conditions of our experiments) and is therefore difficult to define accurately. As a result, we deemed it more suitable to compare our results to published activity measurements derived from 14-3-3-activated dimers which should represent fully dimerized BRAF. This analysis, as reported in Figure 2E, suggests that the BBD activity is approximately half of that of BB.

(3) The 19F-NMR experiments make a good case for broadening of the helix aC signal in the BRAF dimer. From this, the study proposes that after inhibitor binds one subunit, the second unoccupied subunit retains dynamics. It would be useful to address this experimentally, if possible. For example, can the 19F-NMR signal be measured in the presence of inhibitor, to support the prediction that the unoccupied subunit is indeed dynamic and samples multiple conformations as in apo BRAF?

We agree with the reviewer that it would be interesting to determine the dynamic response of BRAF to inhibitor binding. However, this is a challenging undertaking due to the biochemical heterogeneity that occurs at sub saturating inhibitor concentrations. For example, at any given inhibitor concentration, BRAF exists as a mixture of monomers, apo dimers, dimers with one inhibitor molecule, and dimers with two inhibitor molecules bound. This makes it challenging to relate the 19F NMR signal to a single biochemical state. Addressing this would require a substantial experimental effort that we feel is beyond the scope of this study.